# Alteration of Pituitary Tumor Transforming Gene 1 by MicroRNA-186 and 655 Regulates Invasion Ability of Human Oral Squamous Cell Carcinoma

**DOI:** 10.3390/ijms22031021

**Published:** 2021-01-20

**Authors:** Sang Shin Lee, Jong Ho Choi, Seung Mook Lim, Gi Jin Kim, Suk Keun Lee, Yoon Kyung Jeon

**Affiliations:** 1Department of Oral Pathology, College of Dentistry, Gangneung-Wonju National University, Gangneung 25457, Korea; jhchoi@gwnu.ac.kr (J.H.C.); sukkeunlee@hanmail.net (S.K.L.); 2Department of Biomedical Science, CHA University, Seoul 13488, Korea; lsmook17@naver.com (S.M.L.); gjkim@cha.ac.kr (G.J.K.); 3Department of Pathology, Seoul National University College of Medicine, Seoul 03080, Korea

**Keywords:** pituitary tumor-transforming gene 1, oral squamous cell carcinoma, invasion, microRNA

## Abstract

Background: Pituitary tumor-transforming gene 1 (PTTG1) was recently shown to be involved in the progression as well as the metastasis of cancers. However, their expression and function in the invasion of oral squamous cell carcinoma (SCC) remain unclear. Methods: The expressions of PTTG1 and PTTG1-targeted miRNA in oral SCC cell lines and their invasion capability depended on PTTG1 expression were analyzed by quantitative RT-PCR, Western blots, the transwell insert system and Zymography. Results: Invasion abilities were decreased in oral SCC cells treated with siRNA-PTTG1. When PTTG1 were downregulated in oral SCC cells treated with microRNA-186 and -655 inhibited their invasion abilities via MMP-9 activity. Conclusions: These results indicate that alteration of expression of PTTG1 in oral SCC cells by newly identified microRNA-186 and -655 can regulate invasion activity. Therefore, these data offer new insights into further understanding PTTG1 function in oral SCC and should provide new strategies for diagnostic markers for oral SCC.

## 1. Introduction

Oral squamous cell carcinoma (SCC) is one of the most common malignant epithelial oral neoplasms. While the major risk factors for oral SCC are smoking and alcohol, other risk factors include viral infections, mutations, and immune defects [1]. Although oral SCC is based on clinical detection, most cases of oral SCC are not detected until advanced stages because earlier detection is difficult [2]. According to the last GLOBOCAN, worldwide new case and deaths per year of oral cancer, was about 350,000/18,100,000 and 180,000/9,600,027, respectively [3]. Also, the prognosis is very poor and the five-year survival rate of oral SCC is still less than 50%, despite significant improvements in clinical surgery, radiotherapy, and chemotherapy [4]. It is necessary to confirm whether oral SCC is confined locally or has invaded and metastasized to determine appropriate treatments, such as surgery or irradiation [5,6]. Thus, a reliable diagnostic marker for oral SCC isneeded and it is necessary to elucidate the molecular mechanisms involved in oral SCC metastasis. Recently, reports have suggested mining and evaluating molecular diagnostic markers for invasion and metastasis-specific microRNAs of oral SCC [7].

MicroRNAs (miRNAs) are a family of small non-coding (~22-nt long), endogenous, single RNA molecules that control gene expression by binding to the 3′-UTRs of the target mRNAs, regulating cleavage or translational inhibition [8]. Generally, it is well known that one miRNA targets several genes in stage and time-specific manners [9]. These observations suggest that miRNAs play vital roles in numerous biologic processes, including development, tumorigenesis, and metastasis [10]. Thus, studies on target genes associated with oncogenes or tumor suppressors using miRNAs have recently been reported in human cancers [11]. More recently, Tavazoie et al., showed that miRNA-126 and miRNA-335 functioned as metastasis suppressors in human breast cancer, and miRNA-331-3p had suppressive effects on non-small-cell lung carcinoma (NSCLC) invasion and metastasisin vitroandin vivo [12,13]. Recently, Feng et al. reported that miRNA-22 increased by a lentiviral gene delivery system suppressed the cell proliferation, migration, and invasion of oral SCCs through targeting NLRP3, which is an NLR family pyrin domain containing three inflammasomes, in oral SCC tissues and cell lines [14]. In addition, many researchers have reported that several kinds of miRNAs, such as miR-1-3p, miR-9, and miR-30a-5p, controlled the proliferation, invasion, and metastasis of oral SCCs via exchange the expression of various target genes [15,16,17,18]. Interestingly, it was reported that miRNA-186 had a role as a tumor suppressor in oral SCCs and its expression was decreased by inhibition of the signaling activity of ERK and AKT, which are downstream of protein tyrosine phosphatase SHP2 [16,19]. However, whether miRNA-186 regulates oral SCC cell invasion or modulates pituitary tumor-transforming gene (PTTG)1 in oral SCC cells invasion remains poorly understood.

PTTG is an oncogene that was first isolated and identified in a rat pituitary tumor GH4 cell line but not in normal pituitary [20]. PTTGs are classified into PTTG1, PTTG2 and PTTG3, according to the order in which they were identified [21]. Specifically, PTTG1 is a multifunctional protein involved and overexpressed in various endocrine-related cancers, including pituitary, thyroid, breast, ovarian, and uterine tumors [22,23,24]. Previous evidence has shown that it is an important oncoprotein implicated in the progression of several cancer cell types through the metaphase-anaphase transition of the cell cycle [25,26]. However, PTTG1 is highly correlated with tumor invasiveness and is known as a critical gene associated with tumor metastasis, although its expression in normal human tissue is rare [22,27]. Moreover, the downregulation of PTTG1 by small interfering RNA (siRNA) inhibited the migration ability of non-small cell lung cancer (NSCLC) cells through matrix metalloproteinase (MMP) [28]. Specifically, PTTG1 is directly associated with cancer cell migration through MMP-2 [29]. In our previous reports, the regulation of PTTG1 by miRNA-186 correlated with the proliferation and metastasis of trophoblasts, which demonstrate invasion activity during implantation and placental development through the integrin/Rho-family signaling pathway [30]. Nonetheless, the expression and function of PTTG1 in oral SCC, as well as its molecular mechanisms in the metastasis of oral SCC, are unclear and require further investigation.

Therefore, the objectives of the present study were to analyze the expression of PTTG1 in oral SCC cell lines (YD-10B and YD-15) and determine whether their effect on invasion was dependent upon PTTG1 expression using siRNA. In addition, we investigated whether PTTG1-targeted microRNA could regulate the expression and invasion ability of PTTG1 in oral SCC cells.

## 2. Results

### 2.1. Effect of PTTG1 on Oncogenesis of Oral SCC Cell Lines

First, we confirmed the expression of PTTG1 in both oral SCC cell lines YD-10B and YD-15 by qRT-PCR and Western blotting. The expression of PTTG1 mRNA and protein was increased more in the YD-10B cells than in the YD-15 cells (Figure 1A,B). Generally, PTTG1 is involved in oncogenesis [31]. To confirm the effect of PTTG1 on the oral SCC cell lines, we confirmed the viability of the oral SCC cell lines regardless of the PTTG1 expression using PTTG1-specific siRNA treatment (siPTTG1) (Figure 1C). As shown in Figure 1D, the expression of PTTG1 in the YD-10B and YD-15 cell lines was dramatically decreased after siPTTG1 treatment. No expression of p53 was detected in the YD-10B cell line because they have a mutant type of p53, whereas p53 was highly expressed in the YD-15 cells, which have wild-type p53 (Figure 1D). Also, the expression of p53 was decreased in the YD-15 cells by siPTTG1 treatment, but there was no difference in the expression of CK18 between the two cell types (Figure 1D). Furthermore, we confirmed the expression of vimentin because it is overexpressed in most epithelial cancer and correlates with tumor growth and poor prognosis [32]. The expression of vimentin was increased in the YD-10B cells, whereas vimentin expression was decreased in the YD-15 cells compared to the mock control (Figure 1D). However, the morphological characteristics of both cell types were not changed by siPTTG1 treatment. These findings suggest that the expression of PTTG1 affected oncogenesis in the YD-10B and YD-15 cells.

### 2.2. Effect of PTTG1 on Proliferation of YD-10B and YD-15 Cells

Generally, PTTG1 controls cell proliferation by regulating chromatid separation during cell mitosis [26]. Thus, we confirmed the effect of PTTG1 on the proliferation of YD-10B and YD-15 cells by Western blotting and FACS analysis following siPTTG1 treatment. We determined the proportion of cells in S phase in the YD-10B (7.55%) and YD-15 (12.85%) cell lines compared to the mock control group. After siPTTG1 treatment, the proportion of YD-10B cell in S phase was increased (9.18%), whereas it was decreased in YD-15 cells (4%) (Figure 2A). The activation of cyclin D1 is mainly responsible for regulating the G1-S phase transition and useful as prognostic biomarkers in oral SCC [33,34]. These results are consistent with the marked increase in the expression of cyclin D1 protein in YD-10B cells transfected with siPTTG1 compared to the control. In contrast to YD-10B, the expression of cyclin D1 was remarkably decreased in YD-15 cells transfected with siPTTG1 compared to the control (Figure 2B). To analyze cellular proliferation and death, the expression of phosphorylated Akt (p-Akt), which used as a marker for cell proliferation, was increased in both cell lines by treatment with siPTTG1 compared to the control, although no significant difference was detected in cyclin E1, phosphorylated Erk (p-Erk), Bcl-2 or Bak (Figure 2B,C). Interestingly, the expression of phosphorylated mTOR, which used as a marker for autophagy, in the YD-10B cell line was dramatically decreased, but increased in the YD-15 cells, after siPTTG1 treatment. Additionally, the expression of gp130 was increased in the YD-10B cells after siPTTG1 treatment, but decreased in the YD-15 cells, although the expression of STAT3 and phosphorylated STAT3 was decreased (Figure 2D). These results indicate that the knockdown of PTTG1 decreased the proliferation of YD-10B cells and increased YD-15 cell proliferation by upregulating or downregulating the cell cycle, respectively, in a p53 expression-dependent manner.

### 2.3. Effect of PTTG1 on Invasion Ability of YD-10B and YD-15 Cells

Since several groups reported that PTTG1 promoted the invasion ability in different types of cancer cells [28,35], we investigated whether PTTG1 regulated the invasion ability of YD-10B and YD-15 cells using a Transwell chamber assay. The number of invaded YD-10B cells treated with siPTTG1 were significantly decreased compared to the mock control (*p* < 0.05, Figure 3A). Although the invasion ability of the YD-15 cells was lower than that of the YD-10B cells, the number of invaded cells was significantly decreased after siPTTG1 treatment compared to the mock control (*p* < 0.05, Figure 3A). Furthermore, the expression of MMP-2 and MMP-9 released from the YD-10B cells treated with siPTTG1 was markedly decreased compared to the mock control (*p* < 0.05, Figure 3B). In the YD-15B cells, the expression of MMP-9 was significantly decreased but there was no difference in MMP-2 expression following siPTTG1 treatment (*p* < 0.05, Figure 3B). These results suggest that the downregulation of PTTG1 by siPTTG1 treatment suppressed the invasion ability of the YD-10B and YD-15 cells by decreasing the expression of MMPs.

### 2.4. Effect of miR-186 and -655 on Expression of PTTG1 in YD-10B and YD-15 Cells

To associate miRNAs with the regulation of PTTG1 expression, a bioinformatics search was performed in three databases, Microcosm, MicroRNA, and TargetScan, for predicted miRNA-targeting PTTG1 mRNAs. The databases predicted miR-186 and miR-655 as potential miRNAs targeting PTTG1 (Figure 4A). Furthermore, these tools indicated that the mRNA 3′ UTR of PTTG1 matched with miR-186 and miR-655 (Figure 4B). Synthesized miRNA mimics, inhibitor or scrambled control were transfected into the YD-10B and YD-15 cells. In the qRT-PCR analysis, the expression of miR-186 and miR-655 were effectively decreased in YD-10B and YD-15 cells treated with the mimic sequence (*p* < 0.05, Figure 4C,D, respectively), resulting in significantly decreased mRNA and protein expression of PTTG1 compared to the control (*p* < 0.05, miR-186, Figure 4E,G; miR-655, Figure 4F,H). In contrast, YD-10B and YD-15 cells treated with the miR-186 inhibitor sequence showed decreased expression levels, although the decreases were not statically significant (Figure 5A), resulting in increased mRNA and protein expression of PTTG1 compared to the control (*p* < 0.05, Figure 5C,E). As shown in Figure 6B, the expression level of miR-655 decreased in cells treated with the miR-655 inhibitor sequence, resulting in an increased mRNA and protein expression of PTTG1 (Figure 5D,F). However, the miR-655 inhibitor had no impact on PTTG1 protein levels (Figure 5F). These results suggest that the expression of PTTG1 was modulated by miR-186 and miR-655 in the YD-10B and YD-15 cells.

### 2.5. The Expression of PTTG1 Modulated by miR-186 and miR-655 Regulated Invasion Ability of YD-10B and YD-15 Cells

To identify the role of miR-186 and miR-655 in the invasion ability of YD-10B and YD-15 cells, Transwell chamber assays were performed in both cells treated with mimic or inhibitor sequences. The number of YD-10B and YD-15 cells treated with the miR-186 mimic sequence that invaded through the chamber was considerably less compared to the control cells. Quantitative analysis of the cell numbers showed that the YD-10B and YD-15 cells treated with the miR-186 mimic sequence had invasion rates 1.7 or 3.2 times less, respectively, than that of the control cells (*p* < 0.05, Figure 6A). Also, the number of invaded YD-10B and YD-15 cells was 2.6 or 2 times less, respectively, than that of the control cells after treatment with the miR-655 mimic sequence (*p* < 0.05, Figure 6B). Our previous studies showed that MMP-2 and MMP-9 activities can promote oral SCC cell invasion via PTTG1 expression [36]. Considering that, we performed zymography to analyze the activities of MMP-2 and MMP-9 in YD-10B and YD-15 cells treated with or without the miR-186 and miR-655 mimic sequence. The activity of MMP-9 was considerably decreased in YD-10B and YD-15 cells treated with both mimic sequences compared to the control cells. The activity of MMP-2 was decreased in the YD-10B cells treated with the miR-186 mimic sequence. However, that of the YD-15 cells treated with the miR-186 mimic sequence were increased compared to the control cells (Figure 6C). As shown in Figure 6D, the activity of MMP-9 was effectively decreased in the YD-10B and YD-15 cells treated with miR-655 mimic sequence compared to the control cells, although the MMP-2 activities were not significantly different.

In contrast to the mimic sequence results, the number of invaded YD-10B and YD-15 cells treated with the miR-186 or miR-655 inhibitor sequences was higher than that of the control cells. The number of invaded YD-10B and YD-15 cells treated with the miR-186 inhibitor sequence was 1.3 or 1.6 times higher, respectively, than that of the control cells (*p* < 0.05, Figure 7A). In addition, the number of invaded YD-10B and YD-15 cells treated with the miR-655 inhibitor sequence was 1.2 times higher than the control cells (*p* < 0.05, Figure 7B). Furthermore, the activities of MMP-9 were increased in YD-10B and YD-15B cells treated with the inhibitor sequence, regardless of the miRNAs, compared to control cells. However, the MMP-2 activities were not significantly different, regardless of the miRNA treatment (Figure 7C,D). These results strongly suggest that miR-186 and miR-655 were capable of directly targeting the 3′-UTRs of the PTTG1 genes, resulting in the alteration of PTTG1 expression associated with the invasion ability of oral SCC cells via MMP-9 activity.

## 3. Discussion

In the present study, we demonstrated that PTTG1 was expressed in oral SCC cells in a p53-dependent manner that regulated the proliferation of oral SCC cell lines via cell-cycle regulation. In addition, the inhibition of PTTG1 reduced the invasion ability of oral SCC cell lines and their invasion ability was regulated through MMP activity via miR-186 and miR-655.

PTTG1 is a recently identified oncogene in pituitary tumors and high expression levels of PTTG1 were reported in several different cancers [25,37,38]. In the oral SCC field, several groups recently reported the overexpression of PTTG1 in patients with oral SCC and that the expression of PTTG1 was a potential predictor of oral SCC progression. [39,40]. However, the studies did not investigate the mechanisms related to tumorigenesis through PTTG1 and did not confirm a correlation between PTTG1 expression and oral SCC cells in in vitro experiments. In our study, two types of SCC cells originating from oral SCC, YD-10B and YD-15 cells, were used to verify and investigate the mechanism of PTTG1 in oral cancer tumorigenesis.There are differences in the pathologic diagnosis and p53 expression in the two cell lines. The YD-10B cells are oral SCC, which has a frameshift mutation in p53, resulting in no expression of the p53 protein, whereas YD-15 cells strongly express p53 protein [41], which is known as a transcription factor with pro-apoptotic function. Therefore, its expression is associated with tumor suppression by promoting cell cycle arrest and apoptosis [42]. Interestingly, the PTTG1 expression and p53 expression were negative correlation in these cell lines in the present study. These findings are correlated with reports that high expression of PTTG1 was found to be associated with mutant p53 protein stability [43] Additionally, Tong et al. showed that p53 binds to PTTG1 and their interaction was shown to regulate cell cycle regulation via cyclin D using chromatin immunoprecipitation-of-chip studies [44]. Furthermore, several studies reported that p53-mediated PTTG1 expression induced cell death by regulating DNA-damage and/or apoptotic factors [38,45], In our preliminary data, we confirmed the expression patterns of PTTG1 and their localization were observed in oral SCC tissues of patients. Their expression was localized in nuclei of tumor cells in well and moderate-differentiated oral SCC tissues (data not shown). In this study, we showed that the altered expression of PTTG1 regulated the cell cycle of oral SCC cells and their function was involved with p53-dependent cell cycle arrest, although no differences in apoptotic and anti-apoptotic changes were seen. However, these different mechanisms mediated by p53 should be studied in the future.

Recent studies suggested that miRNAs are involved in post-transcriptional gene expression and play critical roles in biological and cellular processes, such as metabolism, tumorigenesis, and metastasis. In cancer studies, miRNA acts as a metastasis regulator, as well as an oncogene and tumor suppressor, depending on the cancer type. MetastamiRs regulate metastasis through diverse mechanisms, including migration, invasion, and epithelial-mesenchymal transition (EMT) [10,46]. To further study the mechanisms of PTTG1 on the invasion ability of oral SCC, we used bioinformatics algorithms and methods, including microRNA [47], TargetScan human [48], and Microcosm Targets [49]. Based on the predictions, we found that miR-186 and -655 might be regulators that interact with PTTG1. Previous studies reported that low plasma levels of miR-655 were associated with lymphatic invasion and lymph node metastasis in patients with esophageal squamous cell carcinoma (ESCC) [50]. The overexpression of miR-655 inhibited the invasion ability of ESCC cells by targeting PTTG1 [51]. Li et al. demonstrated that PTTG1 modulated by miR-186 regulated the invasion and metastasis of NSCLC [28]. Wang et al. showed that PTTG1 was overexpressed in patients with neuroendocrine tumors (CNET), which was regulated by miR-186. However, all previous reports on PTTG1 expression by targeting miR-186 and miR-655 were insufficient to clearly explain the mechanisms of cancer invasion and metastasis. The relationship between miR, including miR-186 and miR-655, and PTTG1, which is involved with the invasion ability of oral SCC, remains unknown although PTTG1-targeting miR-655 inhibit pituitary tumor cell tumorigenesis via p53/PTTG1 regulation feedback loop [52]. In our previous study, we showed that miR-186 directly targeted PTTG1, which controls human trophoblast invasion via integrin and the Rho-family [30]. In the current study, we first demonstrated that miR-186 and miR-655 were direct targets of PTTG1 in oral SCC cell lines. Using Transwell assays, we found that the altered expression of miR-186 and -655 controlled the invasion ability of oral SCC cell lines. These observations may explain roles for miR-186 and -655 in the regulation of oral SCC invasion and metastasis, regardless of cell type. However, the further study on the correlation between microRNAs (e.g., miR-186 and -655) and p53 expression on migration ability of oral SCC should be need in the future.

MMPs are a highly regulated family of zinc-dependent endopeptidases that are associated with tumor progression and metastasis [53,54]. The role of MMPs in cancer invasion is primarily overexpression in invasive malignant tumors [55,56]. In oral SCCs, MMP-1, MMP-8, MMP-10, MMP-12, and MMP-13 are highly expressed in patients with oral SCC [57]. MMP expression may determine the level of metastasis in oral SCC. Makinen et al. demonstrated that the nuclear expression of MMP-13 or the cytoplasmic expression of MMP-2, -8, and -9 was associated with invasion depth and tumor size in oral tongue squamous cell carcinoma [58]. MMP-7 and -9 and MT1-MMP were shown to be closely correlated to invasion depth and progression [59]. In gelatin zymography, we found that the upregulation of miR-186 and miR-655 reduced the activity of MMP-9, whereas downregulated miR-186 and miR-655 induced the activity of MMP-9 in YD-10B and YD-15 cells. These findings suggest that miR-186 and miR-655 negatively controlled the expression of PTTG1 and MMP-9 in the oral SCC cell lines. These data are consistent with previous reports on the PTTG1 control of oral SCC cell lines through the regulation of MMP-2 and MMP-9 activity.

## 4. Materials and Methods

### 4.1. Cell Culture

Two oral SCC cell lines, YD-10B cell line (p53 frameshift mutation, Tongue) and YD-15 cell line (p53 wild type, Tongue) were used [41]. YD-10B and YD-15 were obtained from Dr. Jin Kim (Yonsei University, Seoul, Korea). Both cell lines were cultivated in a defined medium [3:1 mixture ofDulbecco’s modified Eagle’s medium (11965, Gibco, Rockville, MD, USA) and Ham’s F12 nutrient Mixture (11765-054, Gibco) supplemented with 10% fetal bovine serum (16000-044, Gibco) and 100 U/mL penicillin and streptomycin (15140-122, Gibco), and 2 mM L-glutamine (25030, Gibco) at 37 °C in a humidified atmosphere of 5% CO_2_. Synthesized miRNA mimics, inhibitor, and scrambled control were synthesized by Genolution (Genolution Inc., Seoul, Korea). All transient transfections with siRNA (Invitrogen, Waltham, MA, USA) and miRNA (Genolution, Seoul, Korea) in YD-10B and YD-15 cells were performed using Lipofectamine 2000 Transfection Reagent (Invitrogen), according to the manufacturer’s protocol. Briefly, 1 × 10^6^ cells were seeded onto 100 mm culture dishes with serum-free OPTI-MEM (Gibco). Next, the cells were transfected with 200 nM siRNA-PTTG1 (siPTTG1) or 30 nM miRNAs (-186, -655, or scrambled negative control) for 24–48 h. The sequences used in the present study are shown as below. hsa-miR-186-5p; F: 5′- CAA AGA ATT CTC CTT TGG GCT-3′; hsa-miR-655-5p; F: 5′-AGAGGUUAUCCGUGUUAUGUUC-3′.All transfection experiments were optimized for high efficiency and minimal cell damage.

### 4.2. Fluorescence-Activated Cell Sorting Analysis

YD-10B and YD-15 cells were seeded onto 100 mm culture dishes at a density of 1 × 10^6^ cells/dish and were transfected with 50–200nM siPTTG1 for 24 h. The adherent and floating cells were both collected and fixed in ice-cold 70% ethanol at room temperature (RT) for 10 min. The fixed cells were rehydrated in phosphate-buffered saline (PBS) and stained with 50 µg/mLpropidium iodide (Sigma-Aldrich, St. Louis, MO, USA) and 10µg/mLRNase A (Sigma-Aldrich) at room temperature for 30 min. The samples were analyzed by a FACSCalibur flow cytometer (Becton Dickinson, Franklin Lakes, NJ, USA) and the cell cycle phase distribution was quantified with CellQuest Pro software (BD Biosciences, Heidelberg, Germany). All experiments were performed in two separate experiments in duplicate.

### 4.3. Quantitative Real-Time Polymerase Chain Reaction Analysis (qRT-PCR)

Total RNA was extracted using Trizol reagent (Invitrogen). cDNA (500 ng) was synthesized using Superscript III reverse transcriptase (Invitrogen) or a miR-X miRNA First-Strand Synthesis Kit (Clontech Laboratories, Mountain View, CA, USA) according to the manufacturer’s protocol. qRT-PCR was performed on an Exicycler 96 (Bioneer, Deajeon, Korea) using SYBR Green Master Mix (Roche, Basal, Switzerland). The PCR cycling conditions for the mRNAs were as follows: 95 °C for 5 min, followed by 40 cycles at 95 °C for 5 s and 60 °C for 30 s. The microRNA amplification conditions were 95 °C for 10 s, followed by 40 cycles at 95 °C for 5 s and 60 °C for 20 s, with a final step at 95 °C for 60 s, 55 °C for 30 s, and 95 °C for 30 s. All normalizations were done using β-actin or U6 levels and the relative expression levels were calculated by the ΔΔCT method. Also, Reverse transcriptase PCR (RT-PCR) was used to detect mRNA expression of the PTTG1. Total RNA was isolated from the cells using TRIzol reagent (Invitrogen) according to the manufacturer’s instructions. Reverse transcription was carried out using 500ng of total RNA and Superscript III reverse transcriptase (Invitrogen), the synthesized cDNAs were amplified by PCR. The amplification conditions were as follows: 5 min 95°C, followed by 35 cycles of 94 °C for 30 s, 52~65 °C for 1 min, and 72°C for 1 min. The PCR products were visualized by electrophoresis on a 1~2% (*w*/*v*) agarose gel (Cambrex, ME, USA) containing 0.5µg/mLethidium bromide (Promega, WI, USA).The primer sequences used in the present study are shown as the below. PTTG1; F:5′-AAG GAA AAT GGA GAA CA GGC-3′, R: 5′-GCT TGG CTG TTT TTG TTT GAG G-3′; GAPDH; F:5′-GATTCCACCCATGGCAAATTC, R:5′-GTCATGAGTCCTTCCACGATAC.

### 4.4. Western Blots

The cells were extracted on ice in lysis buffer [RIPA buffer (Sigma), complete protease inhibitor cocktail (Mini Tablet; Roche), phosphatase inhibitor cocktail II (A.G. Scientific, San Diego, CA, USA)], and cleared by centrifugation at 13,000 rpm for 20 min at 4 °C. The protein concentration of the lysates was determined using a BCA Protein Assay Kit (Pierce, Rockford, IL, USA). Equal concentrations of total lysates were mixed with 5× loading dye [50mM-Tris HCL(pH 6.8), 2% sodium dodecyl sulfate (SDS), 1% 2-mercaptoethanol, 12.5mM ethylenediaminetetraacetic acid, and 0.02% bromophenol blue], and separated by 8–15% gradient SDS-polyacrylamide gel electrophoresis (PAGE). After electrophoresis, the proteins were transferred to polyvinylidene difluoride membranes (Bio-Rad Laboratories, Berkeley, CA, USA). The membranes were incubated in 8% skim milk or 5% bovine serum albumin (BSA, AMRESCO, Cincinnati, OH, USA) for 1 h at room temperature and then incubated overnight with primary antibodies at 4 °C. The antibodies are shown in Appendix A. The following day, the membranes were incubated with horseradish peroxidase-conjugated anti-rabbit immunoglobulin G (IgG; 1:20,000), anti-mouse IgG (1:20,000) or anti-goat IgG (1:20,000) secondary antibodies for 1 h at room temperature. The specific protein bands were visualized with an Electrochemiluminescence Advance Western Blot Detection Kit (Amersham Biosciences, Uppsala, Sweden).

### 4.5. Transwell Chamber Assay

The invasion ability of oral SCC cells was determined using a 24-well cell culture insert system pre-coated matrigel (8 µm pore size; BD Biosciences) and zymography. For the transwell chamber assay, a total of 3.5–4.5 × 10^4^ cells were collected and added to the upper chamber with serum-free OPTI-MEM media (Gibco). Culture media containing 10% FBS was put into the lower chamber as a chemo-attractant, and the cells were transfected with siRNA or miRNA for 24 h. After 24 h, the membranes were fixed with ice-cold methanol for 20 min and stained with Mayer’s hematoxylin (Dako, Santa Clara, CA USA) at 37 °C for 10 min. The non-invading cells remaining in the upper chamber were scraped out using a cotton swab. The numbers of invaded cells were counted in at least five randomly selected fields under a light microscope. All experiments were performed in two separate experiments in duplicate.

### 4.6. Zymography

After the invasion assay, the supernatants from the invaded cells in the lower chambercontaining secreted MMP-2 and MMP-9 were collected. Equal volumes of the supernatants were electrophoresed on 10% SDS-PAGE gel with 1mg/mLgelatin. The gel was renaturated in renaturation buffer (Bio-Rad Laboratories) at 37 °C for 30 min and incubated in developing buffer (Bio-Rad Laboratories) at 37 °C for 24 h. After incubation, the gel was stained for 2 h with staining solution [10% acetic acid/40% methanol supplemented with 0.5% Coomassie brilliant blue R-250 (Sigma)] and destained for 1 h in destaining solution [50% methanol/10% acetic acid/40%water]. The non-stained bands were used to detect the enzymatic activities of MMP-2 and MMP-9.

### 4.7. Statistical Analysis

The results are presented as means ± standard errors. The data were analyzed using Student’s *t*-test, with *p* < 0.05 considered to indicate statistical significance.

## 5. Conclusions

These findings indicate that PTTG1 is expressed in tumorigenesis of human oral SCC cells, p53 expression dependent manner and the alteration of PTTG1 expression by newly identified microRNA-186 and microRNA-655 can regulate their migration activities. Nevertheless, we need further studies to determine the exact mechanism between PTTG1/p53 or PTTG1/miRNAs in oral SCC. Therefore, these data offer new insight into further understanding the function of PTTG1 in oral SCC and PTTG1-targeting miR-186 and miR-655 as metastatic regulator should provide new strategies to develop more efficient diagnostic markers for oral SCC.

## Figures and Tables

**Figure 1 ijms-22-01021-f001:**
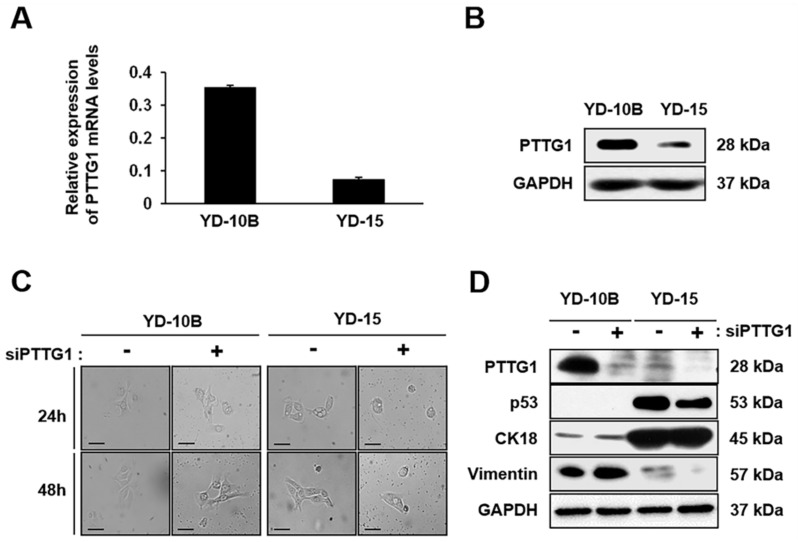
Expression of PTTG1 in oral SCCcell lines. (**A**) The mRNA and (**B**) protein expression of PTTG1 in YD-10B and YD-15 cell lines were analyzed by qRT-PCR and Western blots. (**C**)Morphologies of the YD-10B and YD-15 cell lines after 200 nM siPTTG1 treatment for 24 or 48 h. Magnification, ×400; scale bar = 50 μm. (**D**) The expression of PTTG1, p53, CK18, and vimentin in YD-10B and YD-15 cell lines analyzed by Western blots after 200 nM siPTTG1 treatment for 24 h. GAPDH was used as an internal control. CK18, cytokeratin 18.

**Figure 2 ijms-22-01021-f002:**
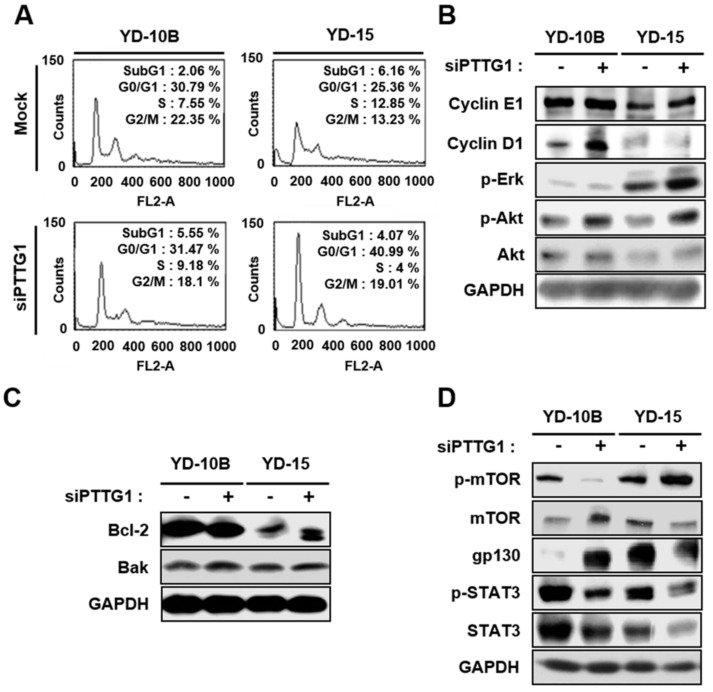
Effect of PTTG1 on cell cycle, apoptosis, and growth in oral SCC celllines. (**A**) Cell cycle analysis by FACScan in YD-10B and YD-15 cell lines after treatment with siPTTG1. The expression of proteins related to (**B**) cell cycle and (**C**) apoptosis, and (**D**) cell growth was analyzed in cells treated with 200 nM siPTTG1 by Western blots. GAPDH was used as an internal control. p-ERK, phosphorylated ERK; p-Akt, phosphorylated Akt; p-mTOR, phosphorylated mTOR; p-STAT3, phosphorylated STAT3.

**Figure 3 ijms-22-01021-f003:**
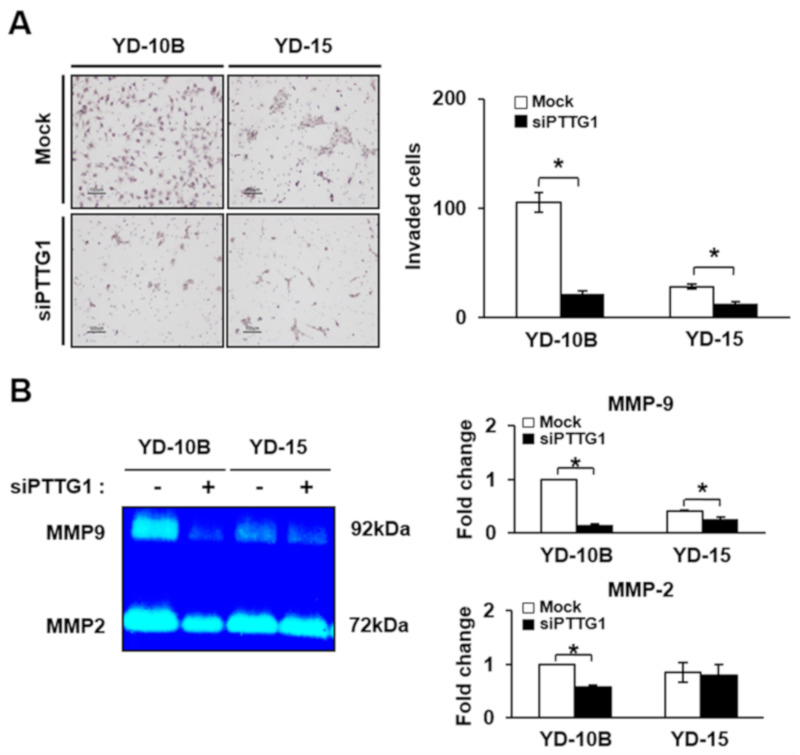
Invasion abilities of oral SCC cell linesdepend on PTTG1 expression. (**A**)Image (left) and numbers (right) of invaded YD-10B and YD-15 cells after siPTTG1 treatment. Magnification, ×100; scale bar = 100 μm. (**B**) Activities of MMP-9 and -2 in YD-10B and YD-15 cells after siPTTG1 treatment analyzed by zymography. The values are means ± standard errors. * indicates significant differences between control and siPTTG1 treatment (*p* < 0.05). All experiments were performed in triplicate.

**Figure 4 ijms-22-01021-f004:**
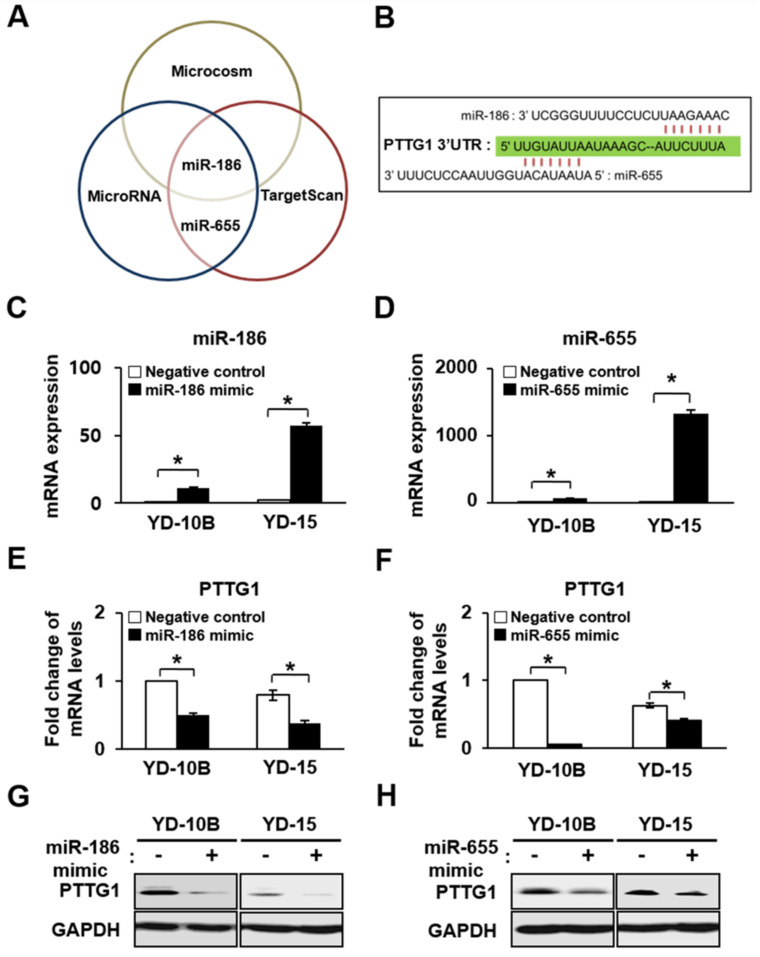
Effect of up-regulated of miR-186 and -655 on PTTG1 expression in oral SCC celllines. (**A**) Venn diagram showing overlapping miRNAs related to PTTG1 mRNA by Microcosm, Micro RNA, and Target Scan. (**B**) miRNA sequence and target of the PTTG1 gene in the 3′-UTR site. Expression of (**C**) miR-186 and (**D**) miR-655 determined by qRT-PCR in YD-10B and YD-15 cells treated with mimic sequences. mRNA expression of PTTG1 determined by qRT-PCR in YD-10B and YD-15 cells treated with (**E**) miR-186 and (**F**) miR-655 mimic sequence. * indicates significant differences between Negative control and mimic treatment (*p* < 0.05). All experiments were performed in triplicate. Protein expression of PTTG1 determined by Western blots in YD-10B and YD-15 cells treated with (**G**) miR-186 and (**H**) miR-655 mimic sequence. GAPDH was used as an internal control.

**Figure 5 ijms-22-01021-f005:**
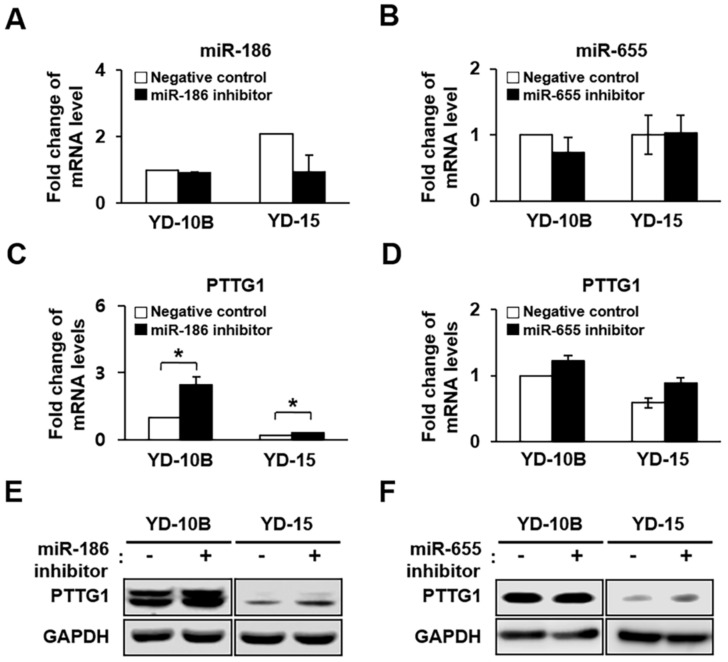
Effect of downregulation of miR-186 and -655 on PTTG1 expression in oral SCC cell lines. Expression of (**A**) miR-186 and (**B**) miR-655 determined by qRT-PCR in YD-10B and YD-15 cells treated with inhibitory sequences. mRNA expression of PTTG1 determined by qRT-PCR in YD-10B and YD-15 cells treated with (**C**) miR-186 and (**D**) miR-655 inhibitory sequences. * indicates significant differences between Negative control and inhibitor treatment (*p* < 0.05). All experiments were performed in triplicate. Protein expression of PTTG1 determined by Western blots in YD-10B and YD-15 cells treated with (**E**) miR-186 and (**F**) miR-655 inhibitory sequence. GAPDH was used as an internal control.

**Figure 6 ijms-22-01021-f006:**
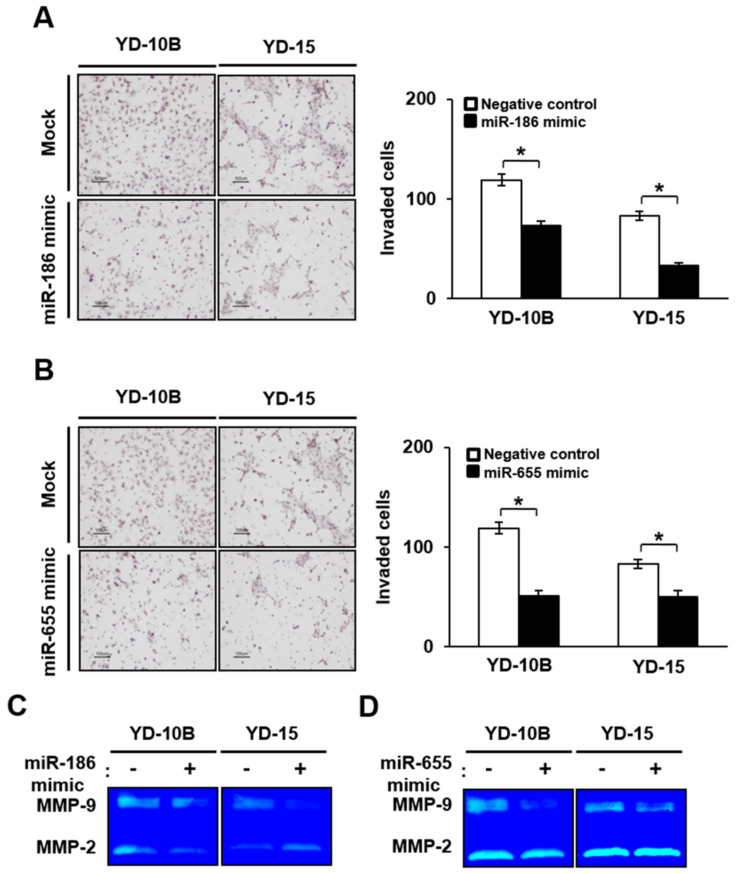
Effect of upregulated miR-186 and -655 on invasion abilities of oral SCCcell lines. Image (left) and numbers (right) of invaded YD-10B and YD-15 cells treated with (**A**) miR-186 and (**B**) miR-655 mimic sequences. Magnification, ×100; scale bar = 100 μm. The values are means ± standard errors.* indicates significant differences between negative control and mimic treatment (*p * < 0.05). All experiments were performed in triplicate. Activity of MMP-9 and -2 analyzed by zymography in YD-10B and YD-15 cells treated with (**C**) miR-186 and (**D**) miR-655 mimic sequences.

**Figure 7 ijms-22-01021-f007:**
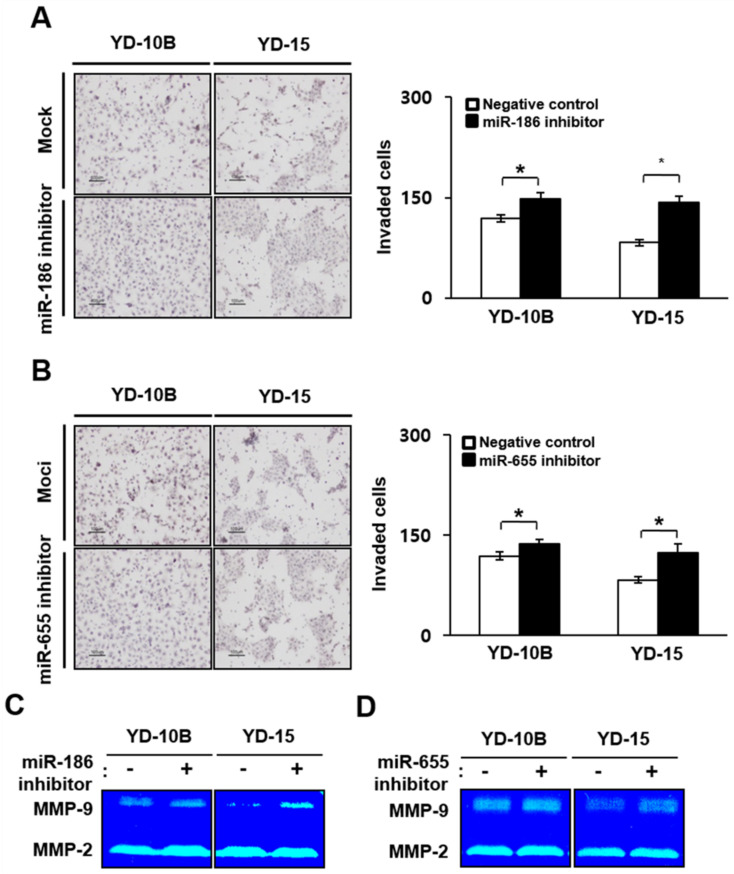
Effect of downregulated miR-186 and -655 on invasion abilities of oral SCC cell lines. Image (left) and numbers (right) of invaded YD-10B and YD-15 cells treated with (**A**) miR-186 and (**B**) miR-655 inhibitory sequences. Magnification, ×100; scale bar = 100 μm. The values are means ± standard errors. * indicates significant differences between negative control and inhibitor treatment. All experiments were performed in triplicate. Activity of MMP-9 and -2 analyzed by zymography in YD-10B and YD-15 cells treated with (**C**) miR-186 and (**D**) miR-655 mimic sequences.

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
