# Peer review of "Alteration of Pituitary Tumor Transforming Gene 1 by MicroRNA-186 and 655 Regulates Invasion Ability of Human Oral Squamous Cell Carcinoma"

_ijms, 2021, doi:10.3390/ijms22031021_

Round 1

Reviewer 1 Report

Although the manuscript submitted for evaluation is interesting and original, certain not very robust aspects should be improved before its consideration for publication. Therefore, I cannot recommend its acceptance in its current state. My following suggestions are given to improve the manuscript:

- Introduction is too short and could be better elaborated.

- A paragraph on oral cancer epidemiology should be included, minimally including number of worldwide new cases and deaths per year (can be extracted from GLOBOCAN last report).

- References are not up to date; I suggest the inclusion of GLOBOCAN last report:

Bray, F., Ferlay, J., Soerjomataram, I., Siegel, R. L., Torre, L. A., & Jemal, A. (2018). Global cancer statistics 2018: GLOBOCAN estimates of incidence and mortality worldwide for 36 cancers in 185 countries. CA: a cancer journal for clinicians, 68(6), 394-424.

- Authors should avoid citing generic bibliography, as there is a specific bibliography for oral cancer. For example, for the oncogenic alteration of cyclin D1, the review of Alao 2007 was cited (Alao, J. P., The regulation of cyclin D1 degradation: roles in cancer development and the potential for therapeutic invention. Mol Cancer 2007, 6, 24.). A better effort searching bibliography should be made, when specific narrative reviews and meta-analyses have been published on cyclin D1 implications in oral oncogenesis:

Ramos-García, P., González-Moles, M. Á., González-Ruiz, L., Ruiz-Ávila, I., Ayén, Á., & Gil-Montoya, J. A. (2018). Prognostic and clinicopathological significance of cyclin D1 expression in oral squamous cell carcinoma: A systematic review and meta-analysis. Oral oncology83, 96–106. https://doi.org/10.1016/j.oraloncology.2018.06.007

Ramos-García P, Gil-Montoya JA, Scully C, Ayén A, González-Ruiz L, Navarro-Triviño FJ, González-Moles MA. An update on the implications of cyclin D1 in oral carcinogenesis. Oral Dis. 2017 Oct;23(7):897-912. doi: 10.1111/odi.12620. Epub 2017 Mar 31. PMID: 27973759.

- The study design was too simple. Only in vitro experiments were performed, on two cancer cell lines (YD-10B and YD-15), not accompanied by experiments in human samples or animal lab experiments. This is an important limitation. Furthermore, nothing was said in relation to the repeatability of experiments, it is not clear whether positive/negative controls were used (only negative controls in some experiments). Finally, experiments were not blinded, a good practice seldom performed in these types of analyses.

The implementation of analyses in human samples using immunohistochemistry, or at least bioinformatics analysis with free datasets obtained from free repositories (for example, Omnibus) would be highly recommended to provide external validation to the experiments carried out here.

Finally, the discussion section could also be improved is a paragraph with limitations is added. High impact journals consider it mandatory. A paragraph including recommendations for future studies would also be highly advised.

Author Response

January 6, 2021

Cover letter

Article Submission in International Journal of Molecular

Dear Editor,

We greatly appreciate your careful evaluation of our manuscript (IJMS-1033486) entitled: “Alteration of pituitary tumor transforming gene 1 by microRNA-186 and 655 regulates invasion ability of human oral squamous cell carcinoma” We were really encouraged by the reviewers’ positive comments and constructive suggestions. I am happy to report that we have successfully addressed all issues and concerns through additional data and subsequent revision of our manuscript, as detailed in the following response page. Changes are highlighted in red in the revised manuscript.

In summary, based on the insightful and constructive criticisms provided by both referees, we believe that our manuscript is significantly improved and we hope that you will consider it suitable for publication in International Journal of Molecular Sciences.

The material contained herein has not been published previously by any of the authors, and is not under consideration for publication in another journal at this time.

Very sincerely yours,

Gi Jin Kim, Ph.D.

Associate Professor

Department of Biomedical Science, CHA University

689, Sampyeong-dong, Bundang-gu, Seongnam-si, Gyeonggi-do, Republic of Korea.

Tel: 82-31-881-3687, Fax: 82-31-881-4102, e-mail: [email protected]

Reviewer 2 Report

“Alteration of pituitary tumor transforming gene 1 by 2 microRNA-186 and 655 regulates invasion ability of 3 human oral squamous cell carcinoma” by Sang Shin Lee et al. is a well-designed and well-written study with a clear conclusion that deserves to be published in IJMS. The authors chose two SCC cell lines, YD-10B with mutated p53 and YD-15 with wt-p53. This model offered identification of differences due to the presence of functional p53. The differences occurred in PTTG1 expression. Then, the inhibition of PTTG1 expression produced a differential response in the abovementioned SCC cell lines in the cell cycle arrest, cyclin D1 and p-mTOR expression. These differences were not discussed and the authors left the space for further research in this direction. However, I suggest discussing why the cell cycle arrest was different in YD-10B and YD-15 as well as why the p-mTOR expression levels were different after PTTG1 inhibition. Association of PTTG1 inhibition with MMP-9 decreased activity and decreased invasion of both cell lines was well-established although a more prominent effect was observed in YD-10B with mutated p53. Even MMP-2 activity was decreased in YD-10B after PTTG1 inhibition. Besides, regulation of PTTG1 expression, MMP-9 activity modulation, and invasion properties of tested cells with miR-186 and miR-665 were determined using miR-mimics and inhibitors.
Minor suggestions:
1. Add western blot and zymography in Methods within the Abstract
2. Discuss the differences due to the presence of functional p53
3. In Figure 5 B and D, write miR-665 inhibitor instead of a miR-186 inhibitor
4. Discuss more truly how these findings can contribute to the better diagnostics and therapy of SCC

Author Response

(The authors gave the same response as above.)

Reviewer 3 Report

“Alteration of pituitary tumor transforming gene 1 by 3 microRNA-186 and 655 regulates invasion ability of 4 human oral squamous cell carcinoma” by Lee et al., describes the role of miRNA 186 and 655 in modulating the PTTG1 in SCC.

  1. To more clearly understand the different cell lines, please add “The YD-10B cells are oral SCC, which has a frameshift mutation in p53, resulting in no expression of the p53 protein, whereas YD-15 cells strongly express p53 protein, in materials and methods, cell lines section.
  2. There are multiple factors involved in metastasis, in this study the two cell lines used behave very differently due to difference in p53 levels. What could be the effect of these miRNAs in a normal oral cell? Authors may also discuss the other factors that inaddition to PTTG1 involved in SCC metastasis?
  3. Fig 1 add the p53 western blot to Fig 1 B. to give an idea how the levels of p53 affect PTTG1.
  4. Fig 1 D is difficult to comprehend, the levels of GAPDH are relatively low , where as the PTTG1 levels are very high in control YD10B, whereas the same in Fig 1 B is relatively low compared to GAPDH, can you quantify the Western blots to better understand the figures?
  5. Same with Fig 2, Western blots need quantification.
  6. Fig 3, B, the fold change for mock YD-15 is less than 0.5-fold, so what is this is compared to? Re draw the graph Mock as 1 and then compare the levels of the siRNA treated.
  7. miR-186 and -655 might be regulators that interact with PTTG1, did you check the homology of these miRNAs to UTR region of PTTG1 (show the start/end codons)? Show at least the % homology.
  8. Fig 4 C, mRNA expression, is it percentage or arbitrary number?
  9. Redraw Fig4 E &F to make the control as 1 and then show the down regulation for each cell lines.
  10. Fig 4. G and H quantify the W blots.
  11. Same comments stand for Fig 5.
  12. Discussion need more information, and provide the rational for using p-ERK, phosphorylated ERK; p-Akt, phosphorylated Akt; p-mTOR, phosphorylated mTOR; p-STAT3, phosphorylated STAT3, etc., in the study.
  13. Even though the study shows that miRNA 186 and 655 have a role in PTTG1 expression and migration of cells, the role of p53 cannot be ruled out as the two cell lines show a difference in behaviors due to the lack of p53, authors to need to explain it in their discussion.
  14. They must confirm that these two miRNAs are not targeting p53.
  15. MMP-9 activity could also be measured using ELISA for quantification purposes.

Author Response

(The authors gave the same response as above.)

Round 2

Reviewer 1 Report

I appreciate the work done by the authors responding to all questions and comments suggested by the colleagues reviewers. Although I think the manuscript is a strong candidate to be published in IJMS, I think the manuscript could be improved even more:

- As can be seen from the manuscript, the authors have responded very slightly to the comments suggested (2 references, and two little paragraphs on discussion section). Please, a better discussion section would be advisable, following both reviewers suggestions.

- In relation to the responses given, all suggested references to cyclin D1 were not included (missing: Ramos-García, P., González-Moles, M. Á ., González-Ruiz, L., Ruiz-Á vila, I., Ayén, Á ., & GilMontoya, J. A. (2018). Prognostic and clinicopathological significance of cyclin D1 expression in oral squamous cell carcinoma: A systematic review and meta-analysis. Oral oncology, 83, 96–106. https://doi.org/10.1016/j.oraloncology.2018.06.007).

Instead, a narrative review was cited (new reference 33), and not the meta-analysis, which provides higher scientific evidence level. I think the authors should cite both.

- Finally, the limitations paragraph was not added. This was a key forgotten/neglected point. Please, be self-critical, high impact journals consider it mandatory. You should for example state in this paragraph all the responses given to reviewer 1’s point 2 (Only in vitro experiments were performed, on two cancer cell lines (YD-10B and YD-15), not accompanied by experiments in human samples or animal lab experiments.). This information (and additional limitations added by authors) should be advisable in a future version of this manuscript.

Author Response

I appreciate the work done by the authors responding to all questions and comments suggested by the colleagues reviewers. Although I think the manuscript is a strong candidate to be published in IJMS, I think the manuscript could be improved even more:

Point #1:
As can be seen from the manuscript, the authors have responded very slightly to the comments suggested (2 references, and two little paragraphs on discussion section). Please, a better discussion section would be advisable, following both reviewers suggestions.

- In relation to the responses given, all suggested references to cyclin D1 were not included (missing: Ramos-García, P., González-Moles, M. Á ., González-Ruiz, L., Ruiz-Á vila, I., Ayén, Á ., & GilMontoya, J. A. (2018). Prognostic and clinicopathological significance of cyclin D1 expression in oral squamous cell carcinoma: A systematic review and meta-analysis. Oral oncology, 83, 96–106. https://doi.org/10.1016/j.oraloncology.2018.06.007).

-Instead, a narrative review was cited (new reference 33), and not the meta-analysis, which provides higher scientific evidence level. I think the authors should cite both.

Author’s response #1:
We appreciate the reviewer’s comment. As you mention, we though that recommended references are appropriate, and add that references in “The activation of cyclin D1 is mainly responsible for regulating the G1-S phase transition and useful as prognostic biomarkers in oral SCC” [See Reference 33 and 34]

Point #2:
Finally, the limitations paragraph was not added. This was a key forgotten/neglected point. Please, be self-critical, high impact journals consider it mandatory.

You should for example state in this paragraph all the responses given to reviewer 1’s point 2 (Only in vitro experiments were performed, on two cancer cell lines (YD-10B and YD-15), not accompanied by experiments in human samples or animal lab experiments.). This information (and additional limitations added by authors) should be advisable in a future version of this manuscript.

Author’s response #2:
Thank you for your comment.

As you mention, we added the limitations paragraph in Discussion part of the revised manuscript about the issue in “In our preliminary data, we confirmed the expression patterns of PTTG1 and their localization were observed in oral SCC tissues of patients. Their expression was localized in nuclei of tumor cells in well and moderate-differentiated oral SCC tissues (data not shown). In this study, we showed that the altered expression of PTTG1 regulated the cell cycle of oral SCC cells and their function was involved with p53-dependent cell cycle arrest, although no differences in apoptotic and anti-apoptotic changes were seen. However, these different mechanisms mediated by p53 should be studied in the future.”

Reviewer 3 Report

Authors have answered most of the queries satisfactorily and the modifications added in the text also improved the quality of the manuscript.

Author Response

Authors have answered most of the queries satisfactorily and the modifications added in the text also improved the quality of the manuscript.

Author’s answer: Thank you for positive response.

Round 3

Reviewer 1 Report

All comments suggested by reviewers were fully taken into account.